# Characterization of the Molecular Diversity and Degranulation Activity of Mastoparan Family Peptides from Wasp Venoms

**DOI:** 10.3390/toxins15050331

**Published:** 2023-05-12

**Authors:** Xiangdong Ye, Xin Liu, Xudong Luo, Fang Sun, Chenhu Qin, Li Ding, Wen Zhu, Huajun Zhang, Haimei Zhou, Zongyun Chen

**Affiliations:** 1Department of Biochemistry and Molecular Biology, Institute of Basic Medical Sciences, College of Basic Medicine, Hubei Key Laboratory of Embryonic Stem Cell Research, Hubei University of Medicine, Shiyan 442000, China; yexiangdong7237@163.com (X.Y.); liuxin00113@126.com (X.L.); luoxudong000@126.com (X.L.); 2016202040055@whu.edu.cn (F.S.); 2016202040025@whu.edu.cn (C.Q.); zhuwen0712@126.com (W.Z.); z1269658612@163.com (H.Z.); zhm030528@126.com (H.Z.); 2Hubei Key Laboratory of Wudang Local Chinese Medicine Research, Hubei University of Medicine, Shiyan 442000, China; 3Department of Clinical Laboratory, Dongfeng Hospital, Hubei University of Medicine, Shiyan 442000, China; dl2168@163.com

**Keywords:** wasp venom, mastoparan, degranulation, structure–function relationship

## Abstract

Wasp stings have become an increasingly serious public health problem because of their high incidence and mortality rates in various countries and regions. Mastoparan family peptides are the most abundant natural peptides in hornet venoms and solitary wasp venom. However, there is a lack of systematic and comprehensive studies on mastoparan family peptides from wasp venoms. In our study, for the first time, we evaluated the molecular diversity of 55 wasp mastoparan family peptides from wasp venoms and divided them into four major subfamilies. Then, we established a wasp peptide library containing all 55 known mastoparan family peptides by chemical synthesis and C-terminal amidation modification, and we systematically evaluated their degranulation activities in two mast cell lines, namely the RBL-2H3 and P815 cell lines. The results showed that among the 55 mastoparans, 35 mastoparans could significantly induce mast cell degranulation, 7 mastoparans had modest mast cell degranulation activity, and 13 mastoparans had little mast cell degranulation activity, suggesting functional variation in mastoparan family peptides from wasp venoms. Structure–function relationship studies found that the composition of amino acids in the hydrophobic face and amidation in the C-terminal region are critical for the degranulation activity of mastoparan family peptides from wasp venoms. Our research will lay a theoretical foundation for studying the mechanism underlying the degranulation activity of wasp mastoparans and provide new evidence to support the molecular design and molecular optimization of natural mastoparan peptides from wasp venoms in the future.

## 1. Introduction

In recent years, wasp stings have become an increasingly serious public health problem because of their high incidence and mortality rates in various countries and regions [1,2,3]. Anaphylaxis is one of the most common and prominent clinical symptoms of wasp stings, the clinical features of which are extensive, from painful local swelling to systemic anaphylactic reactions and shock symptoms. Allergic reactions to wasp venom are also a devastating problem due to progressive immune responses in different systems, including the respiratory system, circulatory system, nervous system, digestive system, blood system, and urinary system, which may lead to multiple organ dysfunction and even death [4,5]. Mast cells (MCs) are innate immune cells that are scattered in tissues throughout the organism and are particularly abundant at sites exposed to the environment, such as the skin and mucosal surfaces. MCs are well known for their pivotal role in allergic responses, where allergens trigger MC activation by crosslinking IgE bound to the high-affinity IgE receptor (FcεRI) leading to the initiation of the allergic cascade. In addition, MCs express a great armamentarium of receptors (such as toll-like receptors and G-protein-coupled receptors) enabling them to respond to a wide variety of cellular, viral, and bacterial triggers [6,7]. Venom allergies may be mediated by both immunological mechanisms and nonimmunological mechanisms [8].

Mastoparan family peptides are the main component of wasp venom, accounting for approximately 50–60% of the dry weight of wasp venom [9], and they were originally described as factors that promote the degranulation of mast cells; these peptides play a pivotal role in allergy reactions and inflammation induced by wasp venom [10]. Some studies found that Mastoparan-L induced the degranulation of connective tissue mast cells (CTMCs) via a G protein-coupled receptor named MRGPRX2 (Mas-related G protein-coupled receptor member X2) and the Gαq/PLCγ1/IP3/Ca^2+^ flux signaling pathway [11,12]. Several researchers have identified many mastoparans in various wasp venoms, most of which have been reported to have different abilities to induce mast cell degranulation [13]. However, the molecular diversity and functional differences of mastoparan family peptides that induce allergic-reaction-mediated organ dysfunction remain unclear.

Here, based on the mastoparan family peptides that have been reported in the previous literature [1,2], and the NCBI PubMed public database, we systematically identified mastoparan family peptides in wasp venoms by bioinformatics methods. Fifty-five mastoparan family peptides derived from 31 wasp species in five wasp families were identified, suggesting the molecular diversity of mastoparan family peptides from wasp venoms. To systematically and comprehensively investigate the degranulation activity of wasp venom mastoparans, we analyzed all the reported mastoparans and established the first wasp mastoparan library by chemical synthesis. We comprehensively evaluated the degranulation activity of 55 mastoparans in RBL-2H3 cells and P815 cells and preliminarily explored the relationship between the structure and mast cell degranulation activity of these mastoparans. Our results will lay a theoretical foundation for research on the mechanism underlying the degranulation activity of wasp mastoparans and provide new evidence to support the molecular design and molecular optimization of natural mastoparan peptides from wasp venoms.

## 2. Results

### 2.1. Molecular Diversity and Physical Properties of Mastoparan Family Peptides from Wasp Venoms

Based on the reported mastoparan peptide sequences and the NCBI PubMed public database, we systematically identified mastoparan family peptides from wasp venoms with bioinformatics methods. Fifty-five mastoparan family peptides derived from 31 wasp species in five wasp families were identified (Table 1), suggesting the molecular diversity of mastoparan family peptides from wasp venoms. As shown in Figure 1A, 13 mastoparans were derived from 7 wasp species of *Eumenidae*, 22 mastoparans were derived from 12 wasp species of *Vespidae*, 9 mastoparans were identified in 5 wasp species of *Polistidae*, 10 mastoparans were identified in 4 wasp species of *Polybiidae*, and only 1 mastoparan was found in *Ropalidia speciosa* of *Ropalidiidae*. Further analysis showed that 17 out of 31 wasp species venom contained only one mastoparan, 8 out of 31 wasp species venom contained two mastoparans, and 4 out of 31 wasp species venom contained three mastoparans. These data indicated that most wasp venom contained only one mastoparan. However, both *Vespa tropica* and *Polybia paulista* contained five mastoparans (Figure 1B).

Among the 55 mastoparan family peptides, 83.6% of mastoparans (46/55) were composed of 14 amino acids, and only 6 mastoparans had 15 amino acids and 2 mastoparans had 17 amino acids (Table 1), which was consistent with previous research that wasp mastoparans were mainly composed of 14 amino acids [9]. Physical property analysis also showed that 54 mastoparans are cationic peptides with one to five positive charges, and only 1 mastoparan is electrically neutral. The amphipathic analysis showed that the mean hydrophobicity of mastoparans was between 0.377 and 0.829, and there were 43 mastoparans with hydrophobic moments between 0.400 and 0.600, 11 mastoparans with hydrophobic moments lower than 0.400, and only 1 with a hydrophobic moment greater than 0.600. As described in previous articles [9], each mastoparan contains two to four lysine residues whose position along the amino acid chain allows the formation of a stable helical structure, which was also consistent with our present analyses. Interestingly, among the 55 mastoparan family peptides, most mastoparans (92.7%, 51/55) had amidation modifications in the C-terminal regions, but there were seven mastoparans without amidation modifications in the C-terminal regions. These results suggest the molecular diversity of mastoparan family peptides from wasp venoms in terms of molecular size, molecular charge, hydrophobicity, hydrophilicity, and C-terminal amidation modification (Table 1).

### 2.2. Classification of Diverse Mastoparan Family Peptides from Wasp Venoms Based on Sequence Alignments

Next, we investigated the relationship of 55 mastoparan family peptides based on their primary structures. The results of the homologous amino acid sequence alignment of 55 mastoparans showed that the N-terminal amino acids of mastoparans are highly conserved, mainly dominated by INW/LK/L (Appendix A and Figure 2A). The C-terminus of mastoparans is less conserved, mainly due to the difference in the number of amino acids. The C-terminal amino acid sequence of mastoparans, which is composed of 14 amino acids, is also conserved and predominantly terminated by amidated leucine (Figure 2A). Based on our sequence alignments, mastoparan family peptides could further be divided into four subfamilies: Subfamily 1 (SF1), Subfamily 2 (SF2), Subfamily 3 (SF3), and Subfamily 4 (SF4) (Figure 2B). The amino acid homology of mastoparans in each subfamily was further analyzed, and the results showed that the amino acid sequences of mastoparan peptides in SF1, SF2, and SF3 are very conserved (Figure 2C–F), and conserved sequence motifs could be found in each subfamily. Mastoparans in SF1 are characterized by the conserved sequence motif INWKKI····K··L, with amino acid difference concentrated in the 7th to 10th and 12th to 13th amino acid residues (Figure 2C). Mastoparan peptides in SF2 are characterized by the sequence motifs IN·K··AA··KK·L (Figure 2D), and amino acid differences are mainly distributed in positions 3, 5, 6, 9, 10, and 13, which is observed throughout the whole polypeptide chain. Mastoparans in SF3 are characterized by the sequence motif INWLKLGK····AL (Figure 2E), and the amino acid differences are mainly concentrated in the 9th to 12th amino acids. The mastoparan peptides in SF4 had the least well-conserved amino acids, and no conserved motif was found in this subfamily (Figure 2F); only the C-terminal amino acids retained some conserved properties. We also observed that some mastoparans that originated from the same wasp venom belong to different subfamilies, such as Pm-R2 belonging to SF2 and Pm-R1/3 belonging to SF3. In conclusion, the known mastoparan family peptides can be divided into four subfamilies. Among them, mastoparan peptides in three subfamilies had conserved motifs, further indicating the molecular diversity and structural conservation of mastoparan family peptides from wasps.

### 2.3. Characterization of the Degranulation Activity of Mastoparan Peptides from Wasp Venoms

Our current work revealed the molecular diversity and classification of mastoparan family peptides from wasp venoms. It is well known that molecular diversity is the foundation of functional differentiation. Mastoparan(-L) was the first cationic α-helical peptide identified in *Vespula lewisii* venom that could induce significant mast cell degranulation; hence, it was also named Mastoparan. Later, multiple studies identified a variety of mastoparans with similar structures and mast cell degranulation activities from different wasp venoms. However, the degranulation effects of mastoparans on mast cells have not been systematically studied. Therefore, it was difficult to analyze the structure and functional relationships of mastoparan family peptides from wasp venoms. To systematically and comprehensively study the influence of different mastoparan peptide families on the detailed functions of these peptides, the degranulation activity of 55 wasp mastoparans was further characterized. First, we established a wasp venom mastoparan library containing all 55 known mastoparan family peptides by chemical synthesis and mass spectrometric identification. Second, we designed a degranulation activity evaluation assay with two representative cell lines, RBL-2H3 cells and P815 cells. RBL-2H3 cells (rat basophil leukemia cells) have been widely used to study the functions of mast cells. We used the 50% effective concentration (EC_50_) and degranulation rate at 80 μg/mL (the lowest dose of mastoparan that reaches 100% degranulation rate) as two indexes to evaluate the degranulation activity of mastoparans in RBL-2H3 cells. The 55 mastoparans were divided into three groups: a high-activity group (EC_50_ ≤ 100 μM), a modest-activity group (100 μM < EC_50_ ≤ 200 μM), and a low-activity group (EC_50_ > 200 μM). As shown in Table 2, there were 35 mastoparans in the high-activity group, which accounted for 63.6% of the mastoparans. Agelaia-MPI, Polybia-MPII, Pm-R3, Protopolybia-MPIII, and EpVP2b induced significant degranulation of RBL-2H3 cells in a dose-dependent manner, and the EC_50_ values were 5.25 ± 0.87 μM, 6.79 ± 0.40 μM, 7.60 ± 0.76 μM, 22.47 ± 2.96 μM, and 25.93 ± 2.88 μM, respectively (Figure 3A–E). Mastoparan(-L) is a widely used agonist for mast cell degranulation, and the EC_50_ was 52.13 ± 3.21 μM (Figure 3F). Seven mastoparans were in the modest-activity group, and 13 mastoparans were in the low-activity group. These results indicated that not all mastoparans have the ability to induce mast cell degranulation and that the activity of different mastoparans varies greatly. Further analysis showed that high-activity mastoparans in *Polistidae* accounted for 88.9% (8/9), which was much higher than the 63.6% (35/55) among all mastoparans, 68.2% (15/22) in *Polybiidae*, 60% (6/10) in *Vespidae*, and 46.15% (6/13) in *Eumenidae* (Figure 4A). Moreover, we investigated the low-activity mastoparans that were identified in five wasp families and found that the low-activity mastoparans identified in *Eumenidae* accounted for 38.5% (5/13), which is much higher than the 23.6% (13/55) among all mastoparans, 31.8% (7/22) in *Vespidae*, 20% (2/10) in *Polybiidae*, and 11.1% (1/9) in *Polistidae*. These results suggest that the number of mastoparans with degranulation activity in *Polistidae* venom is much higher than that in the other four wasp families, while that from *Eumenidae* venom was the lowest. Next, we investigated the degranulation activity of wasp venom peptides in different subfamilies. The results showed that the proportion of high-activity mastoparans in Subfamily 1 (9/11, 81.8%) was much higher than that in SF4 (50%), SF2 (70.6%), and SF3 (71.4%) (Figure 4B).

We also evaluated the degranulation activity of 55 mastoparans in the P815 cell line (mouse mast cell line). The results showed that at the same mastoparan concentration (80 μg/mL), the degranulation activity of all mastoparans in P815 cells was much lower than that in RBL-2H3 cells. However, Agelaia-MPI, Polybia-MPII, Pm-R3, EpVP2b, Protopolybia-MPIII, and Mastoparan-T1 could still induce significant degranulation of P815 cells. These results suggest that the degranulation activity of mastoparans might vary in different cells (Table 2).

### 2.4. Structure–Activity Relationship Studies of Wasp Mastoparan Peptides

Considering the molecular diversity and functional variation of wasp mastoparan family peptides in degranulation activity, we investigated the detailed structure–activity relationship of wasp mastoparans.

In SF1, mastoparan amino acid differences are located at the 7th to 10th and 12th to 13th amino acid residues (Figure 5A) and distributed in both the hydrophilic and hydrophobic surface of α-helical (Figure 5B). We found two pairs of mastoparans with similar amino acid sequences and distinct degranulation activity. The amino acid sequences of Dominulin A and Dominulin B (EC_50_ > 200 μM vs. 94.80 ± 32.77 μM) were different at sites 9, 11, 12, 13, and 16, and those of MP-V1 and MP-V2 (130.24 ± 46.56 μM vs. 88.86 ± 61.29 μM) were different at sites 9 and 15 (Table 2). These results suggest that the amino acid at site 9 of the mastoparans in SF1 may play a pivotal role in degranulation.

In SF2, 12 mastoparans had EC_50_ values lower than 100 μM (red-labeled), and 4 mastoparans had EC_50_ values higher than 200 μM (black-labeled, Figure 6A). The amino acid differences were mainly found in positions 3, 5, 6, 9, 10, and 13, which was observed throughout the whole polypeptide chain. Considering that the secondary structure of mastoparans was a cationic α-helical structure (Figure 6B), we further analyzed the spatial distribution of these amino acid differences in the secondary structure by using the helical wheel. The results showed that the amino acid differences (sites 3, 5, 6, 9, 10, and 13) were all distributed on the hydrophobic surface of the α-helical wheel (Figure 6C,D), which indicated that the hydrophobicity of mastoparans plays a critical role in degranulation activity.

In SF3, 10 mastoparans had EC_50_ values lower than 100 μM. The sequence alignment showed that mastoparans in Subfamily 3 were most conserved, and the amino acid differences were mainly concentrated in the 9th to 12th amino acids (Figure 7A–C). Moreover, in Subfamilies 1–3, the amino acid at site 9 may play a critical role in the degranulation of mast cells.

In addition, although the mastoparans in subfamily 4 are also typical cationic α-helical peptides, we found no significant relationship between their degranulation activities and structures (Figure 8A–C). Interestingly, we found that all mastoparans without amidated modifications (EpVP, Eumenitin, Eumenitin-F, and Eumenitin-R; Table 1) belonged to Subfamily 4, and the EC_50_ of degranulation activity was higher than 100 μM (Table 2); these results mean that the amidation of mastoparans plays an important role in their degranulation activity.

To further explore the amino acids that play a key role in degranulation activity, two mastoparan peptides, Protopolybia-MPIII and Protopolybia-MPI, from the wasp species *Protopolybia exigua* were selected. Both peptides were from Subfamily 3 (SF3), which was the most conserved subfamily, and the peptides had the conserved motif sequence INWLKLGK (Figure 2E). There are only four different residue sites, sites 9 and 11 to 13 (Figure 9A). All the other sites between Protopolybia-MPIII and Protopolybia-MPI were the same. Interestingly, the degranulation activities of Protopolybia-MPIII and Protopolybia-MPI were significantly different. Therefore, we mutated the amino acids at positions 9 and 11–13 of Protopolybia-MPIII to the corresponding sequences of Protopolybia-MPI, and 14 new mutant peptides (Figure 9B) were designed and synthesized with methods similar to those used for the wild-type mastoparan peptides. The degranulation activity of 14 new mutant peptides was further evaluated. The results showed that degranulation activity was significantly decreased after the mutation of Protopolybia-MPIII-9A to 9K (Protopolybia-MPIII-1) and the mutation of Protopolybia-MPIII-13I to 13S (Protopolybia-MPIII-2, -8, -9, -11). When both 9A and 11I were mutated, the degranulation of mast cells was almost lost (Protopolybia-MPIII-5, -13, -14) (Table 3). These results suggest that 9A and 11I are key sites for the mast cell degranulation activity of Protopolybia-MPIII.

## 3. Discussion

Wasp venom is rich in various bioactive substances, such as peptides and proteins [33]. Among these compounds, mastoparans stand out because of their abundance, accounting for 50% of the dry weight of the venom, and their various biological activities, such as their G protein-activating, antimicrobial, anticancer, and hemolytic activities, as well as their mast cell degranulation activity; these features are iconic characteristics of this group of wasp venom peptides [10,13]. Since the discovery of the first mastoparan in wasp venom in 1979 [1], scientists have identified a variety of mastoparans from different wasp venoms. Here, we first studied the molecular diversity of mastoparan family peptides and found that all the known mastoparan family peptides can be divided into four subfamilies: Subfamily 1 (SF1), Subfamily 2 (SF2), Subfamily 3 (SF3), and Subfamily 4 (SF4) (Figure 2). Based on their molecular diversity, we established a polypeptide library containing the 55 wasp mastoparans and systemically evaluated the degranulation activity of the whole family of wasp mastoparans. The ability to induce mast cell degranulation was greatly different among the 55 mastoparans; only 35 mastoparans could induce significant mast cell degranulation (EC_50_ < 100 μM), and 13 mastoparans could hardly induce mast cell degranulation. The degranulation activity of mastoparans varies in different wasp species, such as *Vespa tropicavs*. *Vespa bicolor* (Mastoparan-T4, T2 vs. Mastoparan-VB1, VB2) and *Polistes rothneyiiwataivs*. *Polistes dominulus* (Pm-R1, R3 vs. Dominulin A, B), as well as in different wasp families, such as *Polistidae* vs. *Eumenidae*. The degree of allergic reactions to wasp stings depends on multiple factors, such as the number of stings, the patients’ constitution, and the wasp species (the toxicity of venom) [34]. The venom toxicity of different wasp species has not been reported to date. Our results provide a basis for the preliminary evaluation of the toxicity of different wasp venoms.

The degranulation activity of some mastoparans derived from the same wasp venom differs greatly; for example, Protopolybia-MPI/MPII/MPIII were identified in *Protopolybia exigua*, and the EC_50_ values were 116.76 ± 38.30 μM, 77.85 ± 9.08 μM, and 22.47 ± 2.96 μM, respectively. Pm-R1, Pm-R2, and Pm-R3 were identified in *Polistes rothneyiiwatai*, and the EC_50_ values of Pm-R1, Pm-R2, and Pm-R3 in inducing RBL-2H3 cell degranulation were 22.95 ± 4.77 μM, 94.79 ± 73.52 μM, and 7.60 ± 0.76 μM, respectively (Table 2). Interestingly, the content of Protopolybia-MPIII was significantly higher than that of Protopolybia-MPI/MPII in *Protopolybia exigua* [14], and the contents of Pm-R1 and Pm-R3 were comparable and significantly higher than that of Pm-R2 in *Polistes rothneyiiwatai* venom [15], which suggested the difference in degranulation activity among these mastoparans from the same wasp may be related to their relative contents in wasp venom, and the role of different mastoparans in the same wasp venom played in degranulation activity needs further investigation. In some wasp venom, we also found that not all mastoparans induced mast cell degranulation; for example, there were two mastoparans (Mastoparan-VB1 and Mastoparan-VB2) identified in *Vespa bicolor*, but neither of them could induce the degranulation of RBL-2H3 cells, which suggest that these mastoparans may have other biological activities and need further investigations.

To explore the relationship between mast cell degranulation activity and the structure of mastoparan peptides, two representative mastoparan peptides, Protopolybia-MPIII and Protopolybia-MPI, from the wasp species *Protopolybia exigua* were selected. We found that the C-terminal amino acid residues, especially the 9th to 12th amino acids, play a critical role in mast cell degranulation induced by mastoparans of the SF3. Mutations in both A9K and I11S of Protopolybia-MPIII can significantly decrease degranulation activity. The relationship of mutations in Protopolybia-MPIII and its interactions with intracellular targets needs further investigation. Different subfamilies of mastoparans have different structural characteristics, which might also influence their degranulation activities. For example, amino acid differences were mainly distributed in the hydrophobic surface of α-helices in Subfamily 2 (SF2), which suggests that hydrophobic surfaces may play a pivotal role in inducing mast cell degranulation. However, the relationship between mastoparan hydrophobicity and mast cell degranulation activity has not been reported. Multiple studies have shown that hydrophobicity is critical for multiple biological activities, such as antimicrobial, anticancer, and hemolytic activities [16,17]. The hydrophobic surface of the α-helix can insert into the phospholipid bilayer of cell membranes and generate pores to kill bacteria and some tumor cells. Delazari dos Santos et al. found that Protopolybia-MPIII can cross the cell membrane into the plasma and contribute to the degranulation of mast cells [18], but there is a lack of molecular dynamic analysis for Protopolybia-MPIII and cell membrane interactions, and further investigation is needed.

C-terminal amidation is one of the typical characteristics of wasp mastoparans. In our wasp mastoparan library, four mastoparans were unmodified, among which Eumenitin, Eumenitin-R, and Eumenitin-F hardly induced mast cell degranulation, and EpVP1 had slight degranulation activity (EC_50_ = 171.20 ± 77.20 μM). The relationship between the amidation modification and degranulation activity of mastoparans is still unknown. Ashley L et al. showed that tumor cells (Jurkat T cells) are more sensitive to Mastoparan-NH2 than Mastoparan-COOH, and Mastoparan-COOH had almost no antitumor activity toward Jurkat T cells [19]. Da Silva, A. V. et al. also found that C-terminal amidation promotes the stabilization of the secondary structure of Protonectarina-MP, with a relatively high content of helical conformations, facilitating a deeper interaction with the phospholipid constituents of animal and bacterial cell membranes [35]; this favors biological activities that depend on peptide structure recognition by GPCRs (such as exocytosis) and activities that depend on membrane disruption (such as hemolysis and antibiosis).

Mastoparan(-L) is the most widely studied wasp mastoparan, and multiple studies have shown that mast cell activation induced by Mastoparan(-L) is related to G proteins [21]. Mastoparan(-L) activates mast cells via Mas-related G protein-coupled receptor member X2 (MRGPRX2) and then activates the Gαq/PLCγ1/IP3/Ca^2+^ flux signaling pathway, inducing the degranulation of mast cells [11]. The detailed mechanism underlying the interaction of mastoparan(-L) and MRGPRX2 is still unknown, and no other mastoparans have been found that can activate mast cells via MRGPRX2. Our results showed that the mast cell degranulation activity of 22 mastoparans (EC_50_ > 52.13 ± 3.21 μM) was higher than that of Mastoparan(-L), which suggests that mastoparans may activate mast cells via other more effective pathways. Dos Santos, L. D et al. found that Mastoparan could immediately cross cell membranes and then specifically recognize intracellular proteins and directly activate endosomes, leading to mast cell degranulation [36]. Based on our wasp mastoparan peptide library and the functional data in this current study, it is important to further evaluate the mast cell degranulation induced by wasp mastoparans and explore the pathway and mechanism underlying mast cell activation induced by mastoparans in the future; such studies could provide a theoretical basis for the clinical treatment of wasp-sting-induced allergic reactions and multiple organ dysfunction and the research and development of anti-venom drugs for the treatment of wasp stings.

In conclusion, the amino acids in the C-termini of mastoparans (such as sites 7, 8, 9, 10, 12, and 13 in Subfamily 1 and sites 9, 10, 11, and 12 in Subfamily 3), the composition of amino acids in the hydrophobic surface of the helical wheel structure, and the C-terminal amidation of mastoparans are pivotal for the degranulation activities of wasp mastoparans; these results provide new evidence to support the molecular design and molecular optimization of natural mastoparan peptides from wasp venoms in the future.

## 4. Materials and Methods

### 4.1. Peptide Synthesis and Bioinformatic Analysis

All the peptides were synthesized and purified by ChinaPeptides Corporation (Shanghai, China) by solid-phase methods using standard N-9-fluorenylmethyloxycarbonyl (Fmoc) chemistry [22]. Peptide purity (>95%) was determined by reverse-phase high-performance liquid chromatography (RP-HPLC) with a Kromasil 100-5 C18 column (4.6 mm × 250 mm) at 220 nm at a flow rate of 1.0 mL/min using a linear water/acetonitrile gradient that contained 0.1% trifluoroacetic acid. Peptides were stored as lyophilized powders before use. The theoretical relative molecular mass, isoelectric points (pI), helical-wheel plots, net charge, and hydrophobic moments were calculated online using the HeliQuest server [23]. The conservatism of amino acid sequence of mastoparans using the WebLogo generator (https://weblogo.threeplusone.com/ (accessed on 17 September 2022)) [37]. The PDB structure of mastoparans was obtained with I-TASSER Online Server (https://seq2fun.dcmb.med.umich.edu//I-TASSER/ (accessed on 12 December 2022)) [38]. Multiple amino acid sequence alignments of wasp mastoparans and homology trees were analyzed using DNAMAN software [39]. Fourteen mutations of Protopolybia-MPIII were obtained based on the different amino acids A9K, I11S, D12A, and A13I between Protopolybia-MPIII and Protopolybia-MPI.

### 4.2. Cell Culture

RBL-2H3 cells (rat mucosal-type mast cell line) and P815 cells (mouse mast cell tumor cell line) were obtained from the BeNa Culture Collection (Xinyang, China). RBL-2H3 cells were cultured in DMEM (Gibco, Wuhan, China) supplemented with 15% FBS (Gibco), and P815 cells were maintained in RPMI 1640 basic medium supplemented with 10% FBS and antibiotics (100 mg/L streptomycin and 1 × 10^5^ U/L penicillin) in a humidified atmosphere (5% CO_2_, 37 °C).

### 4.3. Mast Cell Degranulation Activity

Degranulation was analyzed by measuring the release of the granule marker N-acetyl-D-glucosaminidase (β-hexosaminidase), which colocalizes with histamine, as described in [24]. For this, 2 × 10^5^ P815 cells/mL or 1.5 × 10^5^ RBL-2H3 cells/mL were seeded in 96-well cell culture plates. Twelve hours later, the cells were incubated with various concentrations of mastoparans dissolved in 100 μL HEPES buffer (137 mM NaCl, 2.7 mM KCl, 1 mM MgCl_2_, 1.8 mM CaCl_2_, 20 mM HEPES, 1 mg/mL BSA, and 1 mg/mL glucose, pH 7.4) for 15 min at 37 °C and 5% CO_2_. Then, the reactions were quenched by the addition of 0.15 mL of ice-cold HEPES buffer. After centrifugation, the supernatants were sampled for the β-hexosaminidase assay. Briefly, 50 μL samples of the cell culture supernatants and 50 μL of the substrate, 5 mM p-nitrophenyl-N-acetyl-b-D-glucosaminide (Sigma) in 0.2 M citrate, pH 4.5, were incubated in 96-well plates at 37 °C and 5% CO_2_ for 16–18 h to yield the chromophore p-nitrophenol, and the reactions were stopped by the addition of 100 μL of 0.2 M glycine solution, pH 10.7. The absorbance of the colored product was assessed at 405 nm using a microtiter plate reader. The values for β-hexosaminidase released in the medium are expressed as the percentage of total β-hexosaminidase, which was measured in the cells after lysis in 0.1% Triton X-100. The experiment was repeated three times, and the results represent the mean and standard deviation (SD) of quadruplicate tests. The percent degranulation was calculated according to the following equation:Degranulation Rate (%) = 100% × (A_mastoparans_ − A_HEPES_)/(A_Triton X-100_ − A_HEPES_).

### 4.4. Statistical Analysis

Statistical analysis was performed with GraphPad Prism software. The curve of mast cell degranulation induced by wasp mastoparans was generated in SigmaPlot 12.5 software, and the EC_50_ was calculated according to the following equation (Four-Parameter Logistic Curve): y = min + (max − min)/(1 + (x/EC_50_) − Hillslope).

## Figures and Tables

**Figure 1 toxins-15-00331-f001:**
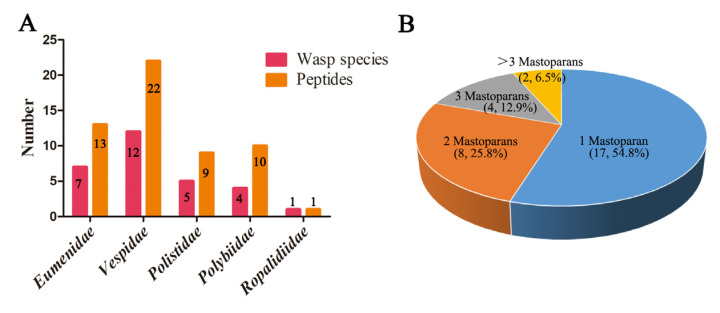
The constitution of mastoparans from different wasp species and families in the wasp mastoparans library. (**A**) The number of mastoparans from wasp species in five wasp families. (**B**) Statistical analysis of the proportion and number of mastoparans from each wasp species.

**Figure 2 toxins-15-00331-f002:**
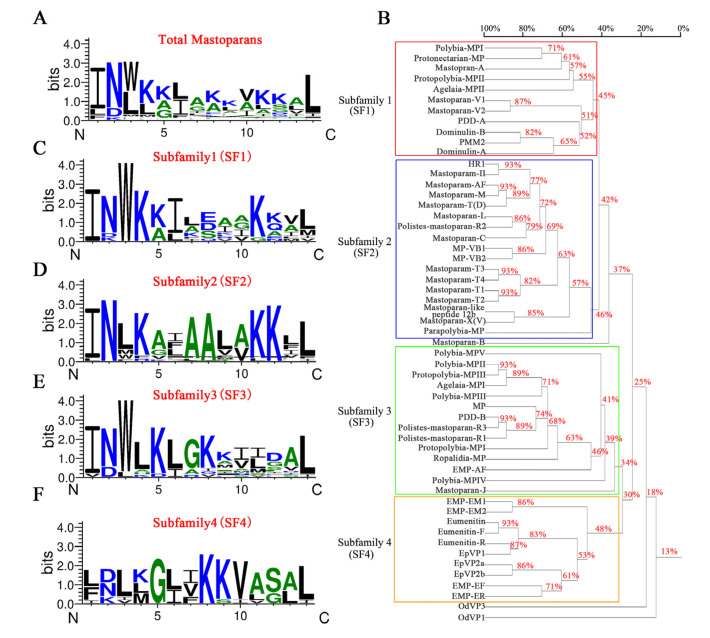
Classification of mastoparan family peptides from wasp venoms according to the different subfamilies. The amino acid sequence alignments of mastoparans using the WebLogo generator. (**A**) The amino acid consensus sequence of 55 mastoparans; (**B**) the subfamily classification of 55 mastoparans; (**C**–**F**) the amino acid consensus sequence of Subfamilies 1, 2, 3, and 4; the overall height in the graph of each stack indicates the sequence conservation at that position (measured in bits), and the height of the symbols within the stack reflects the relative frequency of the corresponding amino acid at that position.

**Figure 3 toxins-15-00331-f003:**
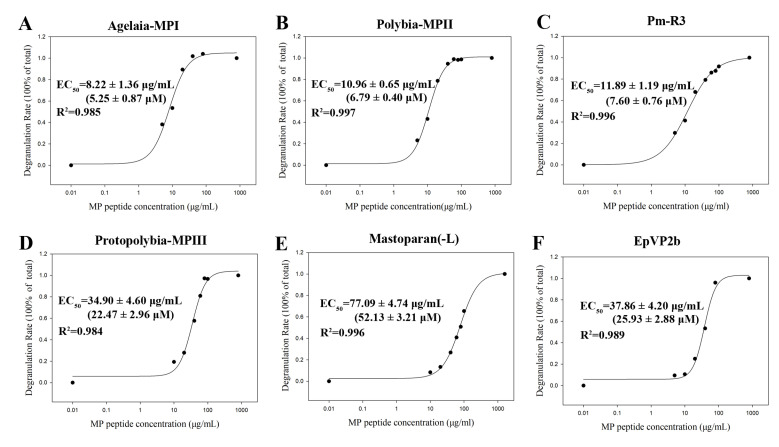
Characterization of the degranulation activity of six representative wasp mastoparan peptides with high activity in RBL-2H3 cells. (**A**–**F**) Curve of RBL-2H3 cell degranulation induced by Agelaia-MPI, Polybia-MPII, Pm-R3, Protopolybia-MPIII, Mastoparan(-L), and EpVP2b. Curves were generated in SigmaPlot 12.5 software, and EC_50_ values were calculated according to the following equation (Four-Parameter Logistic Curve): *y* = min + (max − min)/(1 + (x/EC_50_) − Hillslope).

**Figure 4 toxins-15-00331-f004:**
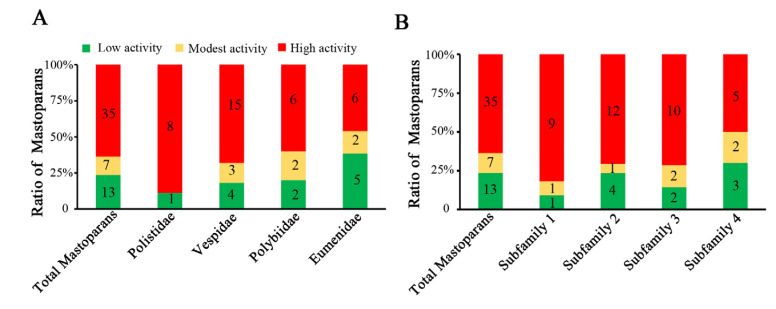
Characterization of degranulation activity and statistical analysis. (**A**) Statistical analysis of the proportion and number of 55 mastoparan family peptides from different wasp species with different degranulation activities in RBL-2H3 cells. (**B**) Statistical analysis of the proportion and number of 55 mastoparan family peptides from different subfamilies.

**Figure 5 toxins-15-00331-f005:**
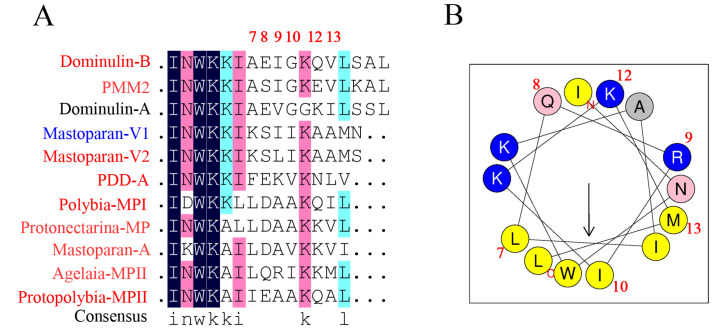
Structure–activity relationship analyses of wasp mastoparan peptides from Subfamily 1 in mast cell degranulation. (**A**) Multiple amino acid sequence alignment analysis of Subfamily 1; mastoparans labeled with red (high degranulation activity), blue (modest degranulation activity), and black (low degranulation activity). Amino acid residues were marked with dark blue (100% identity), pink (>75% identity), light blue (>50% identity), and blank (<50% identity); (**B**) helical wheel projections of a representative of Subfamily 2 (Agelaia-MPII).

**Figure 6 toxins-15-00331-f006:**
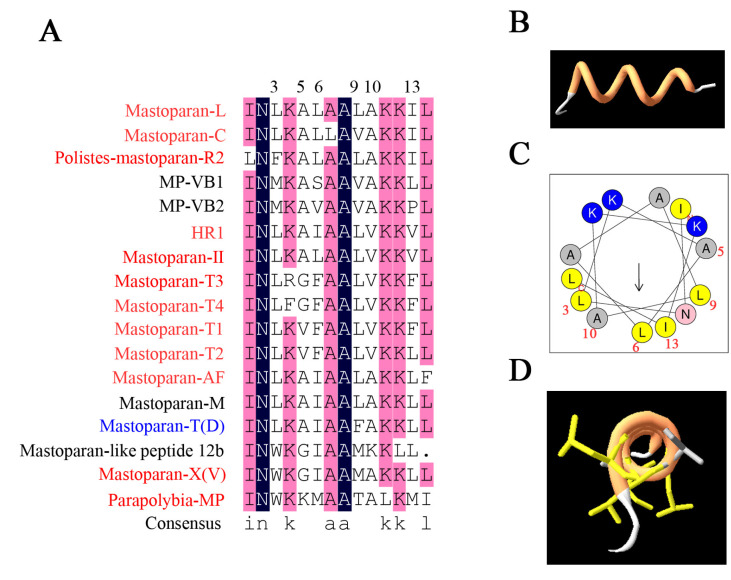
Structure–activity relationship analysis of wasp mastoparan peptides from Subfamily 2 in mast cell degranulation. (**A**) Multiple amino acid sequence alignment analysis of Subfamily 2; mastoparans labeled with red (high degranulation activity), blue (modest degranulation activity), and black (low degranulation activity); amino acid residues were marked with dark blue (100% identity), pink (>75% identity), and blank (<50% identity). (**B**) The exhibition of the PDB structure of Mastoparan L. (**C**) Helical wheel projection of a representative of Subfamily 2 (Mastoparan L). (**D**) The exhibition of side chain of amino acid residues at sites 3, 5, 6, 9, 10, and 13 of Mastoparan (L).

**Figure 7 toxins-15-00331-f007:**
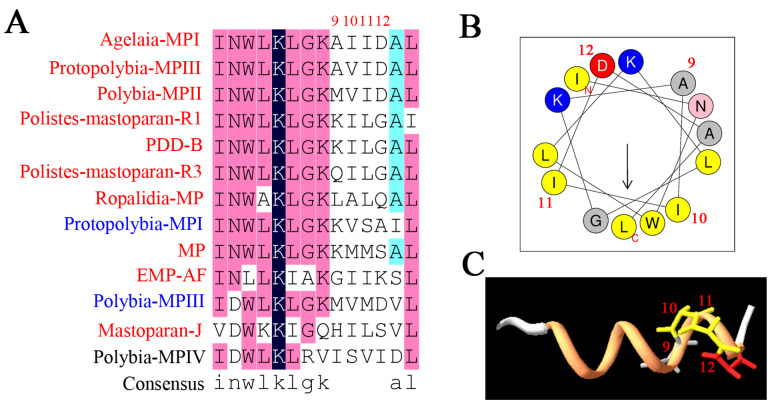
Structure–activity relationship analyses of wasp mastoparan peptides from Subfamily 3 in mast cell degranulation. (**A**) Multiple amino acid sequence alignment analysis of Subfamily 3; mastoparans labeled with red (high degranulation activity), blue (modest degranulation activity), and black (low degranulation activity). Amino acid residues were marked with dark blue (100% identity), pink (>75% identity), light blue (>50% identity), and blank (<50% identity). (**B**) Helical wheel projection of a representative of Subfamily 3 (Agelaia-MPI). (**C**) The exhibition of the PDB structure of Agelaia-MPI.

**Figure 8 toxins-15-00331-f008:**
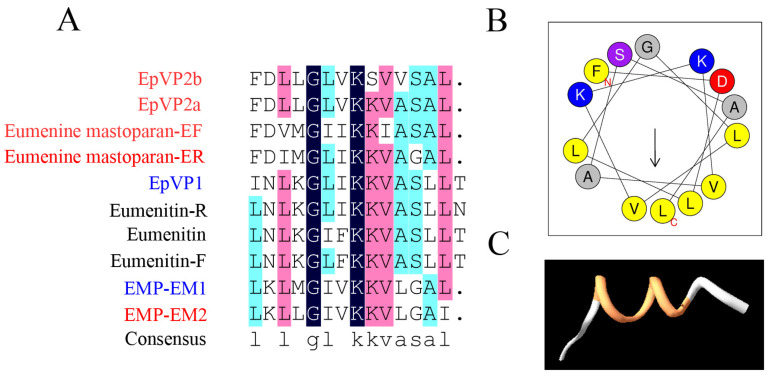
Structure–activity relationship analyses of wasp mastoparan peptides from Subfamily 4 in mast cell degranulation. Multiple amino acid sequence alignment analysis of Subfamily 4; mastoparans labeled with red (high degranulation activity), blue (modest degranulation activity), and black (low degranulation activity); amino acid residues were marked with dark blue (100% identity), pink (>75% identity), light blue (>50% identity), and blank (<50% identity). (**A**) Multiple sequence alignment analysis of Subfamily 4. (**B**) Helical wheel projections of a representative of Subfamily 4 (EpVP2a). (**C**) The exhibition of the PDB structure of EpVP2a.

**Figure 9 toxins-15-00331-f009:**
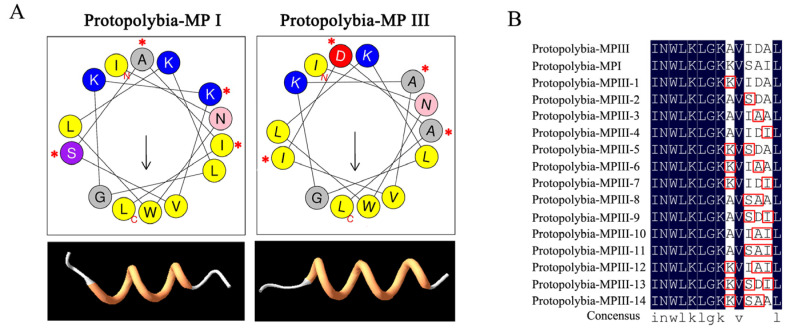
Structure–activity relationship analyses of the effects of two mastoparan wasp peptides, Protopolybia-MPIII and Protopolybia-MPI, on mast cell degranulation. (**A**) The helical wheel projections and PDB structures of Protopolybia-MPIII and Protopolybia-MPI. (**B**) Mutation of a representative member (Protopolybia-MPIII) of Subfamily 3. * and red box: The different amino acids between Protopolybia-MPIII and Protopolybia-MPI.

**Table 1 toxins-15-00331-t001:** The physical property of 55 wasp mastoparans according to different subfamilies and species studied.

Family	Wasp Species	Peptides	Sequence	z ^a^	<H> ^b^	<μH> ^c^	aa ^d^	Mr ^e^	Ref
*Eumenidae*	*Anterhynchium flavormarginatum micado*	EMP-AF	INLLKIAKGIIKSL-NH2	+4	0.643	0.559	14	1522.97	[3]
	*Eumenes fraterculus*	Eumenine mastoparan-EF	FDVMGIIKKIASAL-NH2	+2	0.655	0.489	14	1504.88	[4]
	*Eumenes rubrofemoratus*	Eumenine mastoparan-ER	FDIMGLIKKVAGAL-NH2	+2	0.651	0.493	14	1475.85	[4]
	*Eumenes rubronotatus*	Eumenitin-R	LNLKGLIKKVASLLN *	+3	0.508	0.498	15	1624.04	[4]
		Eumenitin-F	LNLKGLFKKVASLLT *	+3	0.565	0.461	15	1645.06	[4]
		Eumenitin	LNLKGIFKKVASLLT *	+3	0.571	0.465	15	1645.06	[5]
	*Eumenes pomiformis*	EpVP1	INLKGLIKKVASLLT *	+3	0.572	0.455	15	1611.04	[6]
		EpVP2a	FDLLGLVKKVASAL-NH2	+2	0.633	0.457	14	1472.82	[6]
		EpVP2b	FDLLGLVKSVVSAL-NH2	+1	0.766	0.439	14	1459.78	[6]
	*Eumenes micado*	EMP-EM1	LKLMGIVKKVLGAL-NH2	+4	0.686	0.520	14	1481.97	[7]
		EMP-EM2	LKLLGIVKKVLGAI-NH2	+4	0.727	0.540	14	1463.94	[7]
	*Orancistrocerus drewseni*	EMP-OD (OdVP1)	GRILSFIKGLAEHL-NH2	+2	0.589	0.632	14	1552.87	[6]
		OdVP3a	KDLHTVVSAILQAL-NH2	+1	0.595	0.562	14	1506.79	[6]
*Vespidae*	*Vespula lewisii*	Mastoparan (L)	INLKALAALAKKIL-NH2	+4	0.576	0.398	14	1478.91	[8]
	*Vespula vulgaris*	Mastoparan-V1	INWKKIKSIIKAAMN-NH2	+5	0.407	0.428	15	1757.20	[9]
		Mastoparan-V2	INWKKIKSLIKAAMS-NH2	+5	0.437	0.388	15	1730.17	[9]
	*Vespa bicolor*	MP-VB1	INMKASAAVAKKLL-NH2	+4	0.377	0.234	14	1457.84	[10]
		MP-VB2	INMKAVAAVAKKPL-NH2	+4	0.397	0.261	14	1452.85	[10]
	*Vespa xanthoptera*	Mastoparan-X(V)	INWKGIAAMAKKLL-NH2	+4	0.560	0.419	14	1555.97	[11]
	*Vespa analis*	Mastoparan-A	IKWKAILDAVKKVL-NH2	+4	0.541	0.538	14	1624.07	[12]
	*Vespa basalis*	Mastoparan-B	LKLKSIVSWAKKVL-NH2	+5	0.461	0.404	14	1624.07	[12]
	*Vespa crabro*	Mastoparan-C	INLKALLAVAKKIL-NH2	+5	0.641	0.392	14	1506.97	[13]
	*Vespa orientalis*	Mastoparan-II	INLKALAALVKKVL-NH2	+4	0.600	0.416	14	1492.94	[14]
		HR1	INLKAIAALVKKVL-NH2	+4	0.607	0.423	14	1492.94	[15]
	*Vespa tropica*	Mastoparan-T(D)	INLKAIAAFAKKLL-NH2	+4	0.583	0.402	14	1512.93	[16]
		Mastoparan-T1	INLKVFAALVKKFL-NH2	+4	0.712	0.442	14	1603.05	[17]
		Mastoparan-T2	INLKVFAALVKKLL-NH2	+4	0.706	0.436	14	1569.04	[17]
		Mastoparan-T3	INLRGFAALVKKFL-NH2	+4	0.624	0.466	14	1588.99	[17]
		Mastoparan-T4	INLFGFAALVKKFL-NH2	+3	0.824	0.310	14	1579.98	[17]
	*Vespa magnifica*	Mastoparan-like Peptide 12b	INWKGIAAMKKLL-NH2	+4	0.579	0.348	13	1484.89	[18]
	*Vespa mandarinia*	Mastoparan-M	INLKAIAALAKKLL-NH2	+4	0.576	0.399	14	1478.91	[19]
	*Vespa affinis*	Mastoparan AF	INLKAIAALAKKLF-NH2	+4	0.583	0.400	14	1512.93	[12]
	*Agelaia pallipes pallipes*	Agelaia MP-I	INWLKLGKAIIDAL-NH2	+2	0.716	0.538	14	1566.93	[20]
		Agelaia MP-II	INWKAILQRIKKML-NH2	+5	0.556	0.568	14	1754.24	[21]
	*Mischocyttarus phthisicus*	MP	INWLKLGKKMMSAL-NH2	+4	0.594	0.510	14	1632.09	[22]
*Polybiidae*	*Protopolybia exigua*	Protopolybia-MP I	INWLKLGKKVSAIL-NH2	+5	0.634	0.439	14	1581.99	[23]
		Protopolybia-MP II	INWKAIIEAAKQAL-NH2	+2	0.511	0.389	14	1567.88	[23]
		Protopolybia-MP III	INWLKLGKAVIDAL-NH2	+2	0.674	0.506	14	1552.91	[23]
	*Protonectarina sylveirae*	Protonectarina-MP	INWKALLDAAKKVL-NH2	+3	0.497	0.474	14	1581.95	[24]
	*Polybiapaulista*	Polybia-MP I	IDWKKLLDAAKQIL-NH2	+2	0.489	0.511	14	1654.01	[25]
		Polybia-MP II	INWLKLGKMVIDAL-NH2	+2	0.740	0.457	14	1613.02	[26]
		Polybia-MP III	IDWLKLGKMVMDVL-NH2	+1	0.752	0.448	14	1660.09	[27]
		Polybia-MP IV	IDWLKLRVISVIDL-NH2	+1	0.829	0.222	14	1682.07	[28]
		Polybia-MP V	INWHDIAIKNIDAL-NH2	0	0.584	0.420	14	1634.88	[28]
	*Parapolybia indica*	Parapolybia-MP	INWKKMAATALKMI-NH2	+4	0.545	0.412	14	1618.06	[24]
*Polistidae*	*Polistes dorsalis*	PDD-A	INWKKIFQKVKNLV-NH2	+5	0.457	0.565	14	1757.18	[22]
		PDD-B	INWLKLGKKILGAL-NH2	+5	0.671	0.555	14	1565.99	[22]
	*Polistes major major*	PMM2	INWKKIASIGKEVLKAL-NH2	+4	0.450	0.499	17	1910.36	[29]
	*Polistes jadwigae*	Mastoparan-J	VDWKKIGQHILSVL-NH2	+2	0.629	0.519	14	1634.97	[30]
	*Polistes dominulus*	Dominulin A	INWKKIAEVGGKILSSL-NH2	+3	0.488	0.493	17	1855.24	[31]
		Dominulin B	INWKKIAEIGKQVLSAL-NH2	+3	0.495	0.462	17	1910.32	[31]
	*Polistes rothneyi iwatai*	Pm-R1	INWLKLGKKILGAI-NH2	+4	0.678	0.563	14	1565.99	[32]
		Pm-R2	LNFKALAALAKKIL-NH2	+4	0.576	0.408	14	1512.93	[32]
		Pm-R3	INWLKLGKQILGAL-NH2	+3	0.726	0.511	14	1565.95	[32]
*Ropalidiidae*	*Ropalidia*	Ropalidia-MP	INWAKLGKLALQAL-NH2	+3	0.641	0.340	14	1537.90	[23]

a: net charges were measured at pH 7.4; b. *<*H*>*: mean hydrophobicity value represents the sum of all residue hydrophobicity indices divided by the number of residues; c. *<*μH*>*: the hydrophobic moment of each peptide is a value that is relative to the hydrophobic moment of the peptide with perfect amphipathicity; d. aa: amino acids; e. Mr: relative molecular mass of the peptides was measured by mass spectrometry; *. the peptides without amidation at C-terminal.

**Table 2 toxins-15-00331-t002:** The degranulatory activity of 55 mastoparans from wasp venom in RBL-2H3 and P815 cell lines.

Mastoparans	RBL-2H3			RBL-2H3	P815
	EC_50_ (μg/mL)	EC_50_ (μM)	*R* ^2^	Degranulation at 80 μg/mL	Degranulation at 80 μg/mL
Agelaia-MPI	8.22 ± 1.36	5.25 ± 0.87	0.985	100%	68%
Polybia-MPII	10.96 ± 0.65	6.79 ± 0.40	0.997	98%	91%
Pm-R3	11.89 ± 1.19	7.60 ± 0.76	0.996	88%	61%
Mastoparan-AF	27.44 ± 5.95	18.13 ± 3.94	0.986	70%	28%
Mastoparan-T4	28.77 ± 5.55	18.21 ± 3.52	0.984	87%	49%
PDD-B	33.95 ± 5.92	21.68 ± 3.78	0.994	77%	37%
Protopolybia-MPIII	34.90 ± 4.60	22.47 ± 2.96	0.984	97%	56%
Pm-R1	35.94 ± 7.47	22.95 ± 4.77	0.992	76%	33%
EMP-EF	37.80 ± 2.40	25.12 ± 1.59	0.997	82%	39%
EpVP2b	37.86 ± 4.20	25.93 ± 2.88	0.989	96%	65%
Mastoparan-T2	38.02 ± 12.38	24.23 ± 7.89	0.971	76%	29%
Polybia-MPI	38.34 ± 3.46	23.18 ± 2.09	0.997	75%	44%
Mastoparan-A	46.18 ± 2.52	28.43 ± 1.55	0.999	72%	41%
EpVP2a	46.58 ± 6.35	31.63 ± 4.31	0.987	70%	30%
Mastoparan-C	49.19 ± 12.90	32.64 ± 8.56	0.979	58%	35%
Ropalidia-MP	50.68 ± 8.91	32.95 ± 5.79	0.993	68%	21%
Mastoparan-T1	51.48 ± 16.08	32.11 ± 10.03	0.974	64%	58%
Agelaia-MPII	60.88 ± 15.84	34.71 ± 9.03	0.984	59%	27%
Mastoparan-J	60.92 ± 8.39	37.26 ± 5.13	0.985	57%	31%
PMM2	69.93 ± 8.81	36.60 ± 4.61	0.999	55%	22%
HR1	72.86 ± 5.33	48.80 ± 3.57	0.998	52%	32%
Protonectarina-MP	73.43 ± 4.90	46.42 ± 3.10	0.997	52%	30%
Mastoparan(-L)	77.09 ± 4.74	52.13 ± 3.21	0.996	51%	24%
Mastoparan-II	87.62 ± 37.03	58.69 ± 24.80	0.981	50%	28%
EMP-EM2	106.57 ± 28.99	72.80 ± 19.80	0.994	45%	<20%
MP	109.03 ± 50.06	66.80 ± 30.67	0.973	35%	<20%
EMP-ER	112.43 ± 8.58	76.18 ± 5.81	0.999	32%	<20%
PDD-A	115.53 ± 34.31	65.75 ± 19.52	0.992	38%	<20%
EMP-AF	117.13 ± 29.90	76.91 ± 19.64	0.994	39%	<20%
Protopolybia-MPII	122.06 ± 14.24	77.85 ± 9.08	0.999	28%	<20%
Mastoparan-X(V)	122.54 ± 23.30	78.75 ± 14.97	0.997	37%	<20%
Pm-R2	143.4 ± 111.24	94.79 ± 73.52	0.986	43%	<20%
Parapolybia-MP	144.60 ± 32.20	89.36 ± 19.90	0.996	31%	<20%
Dominulin B	148.45 ± 51.31	94.80 ± 32.77	0.996	24%	<20%
MP-V2	153.7 ± 106.0	88.86 ± 61.29	0.976	33%	<20%
Protopolybia-MPI	184.71 ± 60.59	116.76 ± 38.30	0.995	30%	<20%
Polybia-MPIII	188.6 ± 98.37	113.62 ± 59.26	0.988	29%	<20%
Mastoparan-B	203.9 ± 110.9	125.58 ± 68.27	0.989	29%	<20%
Mastoparan-T	209.0 ± 103.6	138.15 ± 68.44	0.991	28%	<20%
MP-V1	228.86 ± 81.81	130.24 ± 46.56	0.998	28%	<20%
EpVP1	275.81 ± 124.37	171.20 ± 77.20	0.996	30%	<20%
MP-EM1	291.47 ± 243.29	196.68 ± 164.2	0.996	44%	<20%
Eumenitin-R	>300	>200	-	14%	<20%
Dominulin A	>300	>200	-	11%	<20%
Mastoparan-like peptide 12b	>300	>200	-	11%	<20%
Mastoparan-M	>300	>200	-	10%	<20%
Eumenitin	>300	>200	-	8%	<20%
Polybia-MPIV	>300	>200	-	6%	<20%
OdVP3	>300	>200	-	5%	<20%
EMP-OD	>300	>200	-	4%	<20%
Polybia-MPV	>300	>200	-	1%	<20%
Mastoparan-VB1	>300	>200	-	−1%	<20%
Mastoparan-VB2	>300	>200	-	0%	<20%
Eumenitin-F	>300	>200	-	2%	<20%

**Table 3 toxins-15-00331-t003:** The statistical analysis of the degranulation activity of Protopolybia-MPIII and mutations.

Mastoparans	Sequences	EC_50_ (μg/mL)	EC_50_ (μM)	*R* ^2^
Protopolybia-MP III	INWLKLGKAVIDAL-NH_2_	33.65 ± 5.34	21.67 ± 3.44	0.9943
Protopolybia-MPI	INWLKLGKKVSAIL-NH_2_	184.71 ± 60.59	116.76 ± 38.30	0.995
Protopolybia-MP III-1	INWLKLGKKVIDAL-NH_2_	158.35 ± 13.57	98.35 ± 8.43	0.9968
Protopolybia-MP III-2	INWLKLGKAVSDAL-NH_2_	294.3 ± 44.02	192.77 ± 28.83	0.9983
Protopolybia-MP III-3	INWLKLGKAVIAAL-NH_2_	23.06 ± 5.33	15.28 ± 3.53	0.9812
Protopolybia-MP III-4	INWLKLGKAVIDIL-NH_2_	44.89 ± 5.92	28.14 ± 3.71	0.9958
Protopolybia-MP III-5	INWLKLGKKVSDAL-NH_2_	>500	>325	-
Protopolybia-MP III-6	INWLKLGKKVIAAL-NH_2_	108.22 ± 19.3	69.11 ± 12.32	0.9887
Protopolybia-MP III-7	INWLKLGKKVIDIL-NH_2_	41.62 ± 2.57	25.19 ± 1.56	0.998
Protopolybia-MP III-8	INWLKLGKAVSAAL-NH_2_	206.95 ± 10.47	139.56 ± 7.06	0.9991
Protopolybia-MP III-9	INWLKLGKAVSDIL-NH_2_	77.82 ± 2.71	49.6 ± 1.73	0.9993
Protopolybia-MP III-10	INWLKLGKAVIAIL-NH_2_	13.02 ± 1.44	8.39 ± 0.93	0.998
Protopolybia-MP III-11	INWLKLGKAVSAIL-NH_2_	>500	>325	-
Protopolybia-MP III-12	INWLKLGKKVIAIL-NH_2_	76.73 ± 22.66	47.72 ± 14.09	0.9584
Protopolybia-MP III-13	INWLKLGKKVSDIL-NH_2_	235.17 ± 39.62	144.63 ± 24.37	0.9953
Protopolybia-MP III-14	INWLKLGKKVSAAL-NH_2_	>500	>325	-

## Data Availability

All data supporting the results can be found within the manuscript.

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
