# Peer review of "Characterization of the Molecular Diversity and Degranulation Activity of Mastoparan Family Peptides from Wasp Venoms"

_toxins, 2023, doi:10.3390/toxins15050331_

Round 1
Reviewer 1 Report
Congratulations, a well done study and a well written manuscript!
I have only a few comments:
- please take care to always write species names in latin (e.g. lanes 70/71)
- there are some typos (e.g. "et al" should be changed to "et al."
- I suggest to include a short section that addresses the biological relevance of the observation: Is there a link between the diversity of mastoparans (both in structure and activity) on one side and the biology/ecology of the respective wasp species (e.g. the number of putative preys or predators) on the other side?
Author Response
Dear editor,
Thanks for you and reviewer’s comments and suggestions for our manuscript “Characterization of the Molecular Diversity and Degranulation Activity of Mastoparan Family Peptides from Wasp Venoms” (Manuscript ID: toxins-2330884). The detail reply to reviewer 1 is as follows:
Reply to reviewer 1
Comments and Suggestions for Authors
Congratulations, a well done study and a well written manuscript!
I have only a few comments:
1、please take care to always write species names in latin (e.g. lanes 70/71)
Reply: we are sorry for our writing mistakes, and we have corrected species names in latin and make sure that all the species names were written correctly.
2、there are some typos (e.g. "et al" should be changed to "et al."
Reply: we are sorry for our writing mistakes, and we have corrected "et al" to "et al." in lines 354, 365 and 380.
3、 I suggest to include a short section that addresses the biological relevance of the observation: Is there a link between the diversity of mastoparans (both in structure and activity) on one side and the biology/ecology of the respective wasp species (e.g. the number of putative preys or predators) on the other side?
Reply: Thanks for reviewer’s advice! it is an important scientific issue, and we have not seen any reports about it. Our present data justly concluded that 80.6% (25/31) wasp species venom contained only one or two mastoparan(s) in Figure 1B, but the detail relationship between the diversity of mastoparans and the biology of wasp species is unclear.
Best Regards.

Reviewer 2 Report
Abstracts needs a short introduction to introduce the subject to a broad scientific community.
For instance, “Wasp stings have become an increasingly serious public health problem being anaphylaxis one of the most common and prominent clinical symptoms. Wasp venom constitutes a devastating problem producing progressive immune responses in different systems, including the respiratory system, circulatory system, nervous system, digestive system, blood system, and urinary system, which may lead to multiple organ dysfunction and even death….”
Authors should explain where Qinba Mountain area is located and other interesting details as it is an important place concerning ecological security, etc.
Authors should explain the criteria for calculating the percentage of degranulation at a peptide concentration of 80 μg/mL.
What kind of statistical analyses have been performed?
Author Response
Dear editor,
Thanks for you and reviewer’s comments and suggestions for our manuscript “Characterization of the Molecular Diversity and Degranulation Activity of Mastoparan Family Peptides from Wasp Venoms” (Manuscript ID: toxins-2330884). The detail reply to reviewer 2 is as follows:
Reply to reviewer 2
Comments and Suggestions for Authors
1、Abstracts needs a short introduction to introduce the subject to a broad scientific community.
For instance, “Wasp stings have become an increasingly serious public health problem being anaphylaxis one of the most common and prominent clinical symptoms. Wasp venom constitutes a devastating problem producing progressive immune responses in different systems, including the respiratory system, circulatory system, nervous system, digestive system, blood system, and urinary system, which may lead to multiple organ dysfunction and even death….”
Reply: Thanks for reviewer’s good advice, and we have added a short introduction in line 4 and 5 marked with green colour.
2、Authors should explain where Qinba Mountain area is located and other interesting details as it is an important place concerning ecological security, etc.
Reply: Thanks for reviewer’s good advice, Qinba Mountain area is located in the middle of China where is one of the highest incidences of wasp stings in China. In fact, the results of transcriptome sequencing of venom glands of Vespa basalis and Vespa nigrithorax in the Qinba Mountain area are not needed (not used) in the present work, so, we changed to “based on the mastoparan family peptides that have been reported in the previous literatures” in lines 51 and 66 labeled by green colour.
3、Authors should explain the criteria for calculating the percentage of degranulation at a peptide concentration of 80 μg/mL.
Reply: Thank you for your professional question! In the preliminary experiment, we found that Agelaia-MPI had the highest degranulation activity (EC50=8.22±1.36 μg/mL / 5.25±0.87 μM), reaching 100% at 80 μg/mL, therefore, we showed the degranulation rate of each mastoparan at 80 μg/mL.
4、What kind of statistical analyses have been performed?
Reply: Thank you! We have not analysed the significance of degranulation activity between different mastoparans in this paper. In our results, the degranulation rate of each dose of mastoparan was obtained by three biological replicates, and the average values were used to calculated the EC50 and curve of degranulation, which was descripted in line 437 to 438.
Best Regards.

Reviewer 3 Report
Reference: “Characterization of the Molecular Diversity and Degranulation Activity of Mastoparan Family Peptides from Wasp Venoms” Submitted to TOXINS, March, 2023.
General comments: In this manuscript, the authors studied the family of peptides called Mastoparan found in hornet and solitary wasp venom. Evaluations of the molecular diversity of these peptides were carried out, and 55 members were described, which were subdivided into 4 subclasses. All peptides were chemically synthesized and had their mast cell degranulating activities evaluated in 2 cell lines. The data showed that among these molecules, 35 peptides showed high mast cell degranulating activity. Through structural studies, the authors showed that the C-terminal amidation and a hydrophobic region of the peptides are important for the degranulatory activity of mast cells. After careful reading of the manuscript it is my opinion that the text has the potential to be published in TOXINS. The work has good methodological criteria, is well written and substantiated and its subject fits the topic of TOXINS. Here are some suggestions and doubts that the authors can forward to the Editorial Board in a revised version.
Specific Comments:
1- The text shown between lines 22 and 29 can be output, as it has already been shown throughout the abstract section.
2- The sentence between lines 42 and 43 could be more detailed.
3- Introduction was very well written, concise, clear, comprehensive and preparatory for the next chapter of results.
4- To increase the readers' attention and make the text more competitive, I would place a figure right after the end of the introduction showing specimens of the main studied wasps (5 Families). This could draw the attention of clinicians involved in the care of injured patients.
5- Between lines 70 to 73 the authors wrote …. Based on transcriptome sequencing of venom glands of Vespa basalis and Vespa nig rithorax in the Qinba Mountain area, and the NCBI PubMed public database, we systematically identified mastoparan family peptides from wasp venoms with bioinformatics methods. It is unclear whether the annotated transcriptome analysis was done for this manuscript, or was for another work. In the first case the data should be part of this manuscript including results and material and methods, and in the second the reference should be cited.
6- In my opinion, the title of Table 1 could include, in addition to the physical properties, also the sub-families and species studied. Something like: The physical property of 55 wasp mastoparans according to different sub-families and species studied.
7- Between lines 145 and 146 and 398…. please replace molecular weight by molecular mass. Although the literature accepts the term molecular weight, this is wrong. The correct one is molecular mass. Just for example, there is mass spectrometry and not weight spectrometry.
8- In the line 85 please correct and separate the words …. Mastoparanshad15 amino acids
9- The same for line 96 …. among the55 mastoparan
10- Also why not place Table 1 before figure 2 in the text. It seems clear to me that Table 1 complements the data shown in figure 1 and thus more logical data would be shown.
11- About figure 2, line 136, I would change title. …from Subfamily classification of mastoparan family peptides from wasp venoms … to …. Classification of mastoparan family peptides from wasp venoms according different subfamilies.
12- In the lines 136 and 137 … The sequence alignments of mastoparans… I would put …the amino acid sequence alignments …to differentiate from nucleotides alignments!
13- Still regarding Figure 2, the authors forgot to write the meaning of the different colors for the amino residues for letters A, C, D, E, and F. Would it be their classifications according to polarity, charges and hydrophobicity ?
14- In the line 152 the authors wrote …. and some scientists called it Mastoparan-L. Do the authors know why this molecule is called mastoparan L? why the letter L? an explanation for this could be write in the text.
15- The sentence between lines 176 to 179 is repetitive and has been written previously, may be removed, or is incomplete in this part of the text. …..Fifty-five mastoparans were identified in five wasp families, including Eumenidae (13 mastoparans), Vespidae (22 mastoparans), Polistidae (9 mastoparans), Polybiidae (10 mastoparans) and Ropalidiidae (1 mastoparan).
16- Between lines 185 to 187 the authors wrote …. These results suggest that Polistidae venom may be more allergically reactive and toxic than venom from the other four wasp families, while Eumenidae venom may be the least toxic. In my opinion the authors have data to discuss mast cell degranulating activities arising from mastoparans. But there are other toxins in the venoms and this conclusion may be speculative in terms of the allergenic potential of the studied species. Also this sentence would look better in the discussion.
17- In the legend of figure 3, Line 194, author could invert first Mastoporan (L) and then EpVP2b accordingly to the figures shown or invert the figures E and F.
18- Regarding Table 2, would it be good if the authors explained why they tested mast cell degranulation at 80 μg/mL? Any reason for not to test 100 μg/mL or 50 μg/mL?
19- Also on Table 2, as shown, it seems that the EC50 (μg/mL,) EC50 (μM) and R2 data were obtained from analyzes with RBL-2H3 cells. Is this interpretation correct? I think the table could be improved and clearer about this.
20- I also would change the title of Table 2 . from … The statistical of degranulation activity of 55 mastoparans in RBL-2H3 and P815 cells ….to … The degranulatory activity of mastoparans from wasp venoms on RBL-2H3 and P815 mast cell lines.
21- Still on Table 2, in my opinion it should be shown in the text before figures 3 and 4, since it describes the same data shown for some mastoparans in figure 3, and additionally, figure 4 is calculated from data obtained in Table 2.
22- Between lines 209 to 213 …. The difference in degranulation activity among these mastoparans from the same wasp may be related to their relative contents in wasp venom. For example, the content of Protopolybia-MPIII was significantly higher than that of Protopolybia-MPI/MPII in Protopolybia exigua[14], and the contents of Pm-R1 and Pm-R3 were comparable and significantly higher than that of Pm-R2 in Polistes rothneyiiwatai venom[15]. Two possibilities: 1- this interpretation should come in the discussion. 2- more or less concentrated toxins in different venoms does not mean that these toxins are more or less active. They can be part of a family that act synergistically or summarily in the same biological event.
23- Lines 213 to 216 …. These results suggest that one or two mastoparans in wasp venom may be the main active ingredient(s) that induce(s) mast cell degranulation, and other mastoparans may have different biological functions. Again in my opinion this interpretation of the results should come in the discussion. Also this sentence is controversial since the actions can be summative, different members of a Family of a same toxin in the same venom can also mean that this venom is adapted to act in several cellular conditions, different membranes, different pHs, humidity, resistance to inhibitors, among other explanations. I suggest to authors to complete this sentence!
24- Lines 216 to 219 … In some wasp venom, we also found that not all mastoparans induced mast cell degranulation; for example, there were two mastoparans (Mastoparan-VB1 and Mastoparan-VB2) identified in Vespa bicolor, but neither of them could induce the degranulation of RBL-2H3 cells, which may be related to the toxicity of the wasps themselves. Once again, more suitable phrase for the discussion of the text. In addition, not causing mast cell degranulation does not mean absence of activities of these molecules. These mastoparans can bind to mast cells and cause activation of intracellular signals responsible for the release of chemokines, among several other signals. Here additional experiments could be done as an immunofluorescence reaction, to show whether these toxins bind to the surface of the cells, originating, for example, formation of microvesicles on the surface or diverse intracellular signals.
25- Lines 225 to 226 …The authors wrote …. These results suggest that the mechanism underlying mast cell degranulation induced by mastoparans from different species may be different (Table 2). The degranulation mechanisms may be the same! What can vary are the receptors for the peptides, which can be expressed at different concentrations. Also the lipid composition of the cell membrane in different cells is different, thus influencing the same intracellular signaling. In addition to the cytoskeleton, inactive genes, among other examples of the intracellular machine that can participate in the same cell signaling. I suggest for the authors complete this phrase!
26- Regarding the data shown in figures 5, 6, 7 and 8 since the authors have all the mastoparans synthesized and purified, it would be interesting to carry out NMR analyses, and to determine the real structural organizations, if not all, of at least some of these peptides. This would be more reliable than performing comparative analyzes with mastoparan (L).
27- For legends of figures 5, 6, 7 and 8 complete…. Multiple amino acid sequence alignment analysis…..
28- Also in the analysis of amino acid alignment in figures 5, 6, 7 and 8 the amino acid residues were marked with different colors (black, pink, white and green), some special reason for this since there is no information in the text or subtitles?
29- Even through comparative analyses, using mastoparan (L) as a reference, the authors could carry out molecular dynamic analyses, which could bring more reliability to the hypotheses raised for the structure/function relationships.
30- The data shown in Table 3 and figure 9 are quite interesting and bring competitiveness to the text. I only suggest for other studies, that the authors carry out molecular dynamic studies comparing the wild forms with the mutated ones, using models of artificial membranes for instance. These data may strengthen the hypotheses indicated by the mutations.
31- The text written between lines 308 to 313 could come at the end of the discussion as it is interpretive and conclusive.
32- In the line 315 ….Wasp venom is rich in various bioactive substances, such as peptides and proteins. Please cite a reference after the text.
33- In the lines 320 and 321 ….Since the discovery of the first mastoparan in wasp venom in 1979, scientists have identified a variety of mastoparans from different wasp venoms. Please cite a reference after the text.
34- In the lines 333 to 335 …. The degree of allergic reactions to wasp stings depends on multiple factors, such as the number of stings, the patients’ constitution and the wasp species (the toxicity of venom). Please cite a reference after the text.
35- In the lines 345 to 346 … In addition, 55 mastoparans can be divided into 4 subfamilies. This information was already written in other part of the text. Please remove it from here.
36- The text in the lines 355 to 357 … Delazari dos Santos et al found that Protopolybia-MPIII can cross the cell membrane into the plasma and contribute to the degranulation of mast cells [18], but the role of mastoparan hydrophobicity in membrane crossing needs further investigation. This part of the text is a clear justification to development of Molecular Dynamic analysis for mastoparans and cell membrane interactions!
37- In the line 365 the authors wrote …. Alessadra V et al. also found that …. The right citation for reference 20 is Cerovsky et al., Please check all references indicated along the text to avoid mistakes.
38- In the line 380 the authors wrote … Delazari dos Santos L et al … The right citation for reference 18 is Goncalves et al.,
39- In the lines 400 and 401 … Multiple alignment of wasp mastoparans amino acid sequences and homology trees were analyzed using DNAMAN software. Please indicate a reference for this methodology.
40- Line 400…. Multiple amino acid alignment …. Include the word amino acid
41- Authors must include in the Material and Methods the details of how methodology they used to obtain mutated isoforms of mastoparans.
42- Also transcriptome analysis.
43- Throughout the text, the authors describe the degranulatory potentials of different mastoparans from wasp venoms on two mast cell lineages, under culture conditions. I ask if in any situation, animal mast cells were tested, for example from the mesentery of mice. This is a simple procedure where the mesentery of mice is collected and stretched over a plate with paraffin for support, and then exposed to the toxins studied at different concentrations and times, and then the mesentery is stained with toluidine blue and observed under light microscopy. Alternatively, animals could be treated in vivo and collected their mesentery to study mast cell degranulation.
44- Of course mastoparans were initially studied as mast cell degranulators. But I missed analyzes on other cellular events, such as the inflammatory response in animals, or effects on other cells co-resident with mast cells, such as fibroblasts and endothelium of blood vessels. These could, although for other manuscripts, bring great improving in the area.
45- Finally, in the legend of supplementary figure 1, the authors should indicate the meanings of the green and pink colors in the amino acid residues.
Author Response
Dear editor,
Thanks for you and reviewer’s comments and suggestions for our manuscript “Characterization of the Molecular Diversity and Degranulation Activity of Mastoparan Family Peptides from Wasp Venoms” (Manuscript ID: toxins-2330884). The detail reply to reviewer 3 is as follows:
Reply to reviewer 3
Comments and Suggestions for Authors
Reference: “Characterization of the Molecular Diversity and Degranulation Activity of Mastoparan Family Peptides from Wasp Venoms” Submitted to TOXINS, March, 2023.
General comments: In this manuscript, the authors studied the family of peptides called Mastoparan found in hornet and solitary wasp venom. Evaluations of the molecular diversity of these peptides were carried out, and 55 members were described, which were subdivided into 4 subclasses. All peptides were chemically synthesized and had their mast cell degranulating activities evaluated in 2 cell lines. The data showed that among these molecules, 35 peptides showed high mast cell degranulating activity. Through structural studies, the authors showed that the C-terminal amidation and a hydrophobic region of the peptides are important for the degranulatory activity of mast cells. After careful reading of the manuscript it is my opinion that the text has the potential to be published in TOXINS. The work has good methodological criteria, is well written and substantiated and its subject fits the topic of TOXINS. Here are some suggestions and doubts that the authors can forward to the Editorial Board in a revised version.
Specific Comments:
1-The text shown between lines 22 and 29 can be output, as it has already been shown throughout the abstract section.
Reply: Thanks for reviewer’s good advice, we have deleted lines 22 to 29 in revised manuscript.
2-The sentence between lines 42 and 43 could be more detailed.
Reply: Thanks for reviewer’s good advice, we have changed it to “MCs are well known for their pivotal role in allergic responses, where allergens trigger MC activation by crosslinking IgE bound to the high affinity IgE receptor (FcεRI) leading to initiation of the allergic cascade. In addition, MCs also express a great armamentarium of receptors (such as toll-like receptors and G-protein-coupled receptors) enabling them to respond to a wide variety of cellular, viral, and bacterial triggers.” in line 35 to 38 labeled by green color in revised manuscript.
3-Introduction was very well written, concise, clear, comprehensive and preparatory for the next chapter of results.
Reply: Thanks for reviewer’s appreciation.
4-To increase the readers' attention and make the text more competitive, I would place a figure right after the end of the introduction showing specimens of the main studied wasps (5 Families). This could draw the attention of clinicians involved in the care of injured patients.
Reply: Thanks for reviewer’s good advice, we are sorry for that we have no specimens of the main studied wasps. Considering that quoting images may involve copyright disputes, therefore we did not place a picture figure in our manuscript.
5-Between lines 70 to 73 the authors wrote …. Based on transcriptome sequencing of venom glands of Vespa basalis and Vespa nig rithorax in the Qinba Mountain area, and the NCBI PubMed public database, we systematically identified mastoparan family peptides from wasp venoms with bioinformatics methods. It is unclear whether the annotated transcriptome analysis was done for this manuscript, or was for another work. In the first case the data should be part of this manuscript including results and material and methods, and in the second the reference should be cited.
Reply: Thanks for your kindly suggestions! Considering that we did not use the data of the “transcriptome sequencing of venom glands of Vespa basalis and Vespa nigrithorax in the Qinba Mountain area” and we also have plan to publish the data as a separate paper in the future, we removed the description of “transcriptome sequencing of venom glands of Vespa basalis and Vespa nigrithorax in the Qinba Mountain area”. In the revised manuscript, we described the section as “based on the mastoparan family peptides that have been reported in the previous literatures [1,2], and the NCBI PubMed public database, we systematically identified mastoparan family peptides in wasp venoms by bioinformatics methods”.
6-In my opinion, the title of Table 1 could include, in addition to the physical properties, also the sub-families and species studied. Something like: The physical property of 55 wasp mastoparans according to different sub-families and species studied.
Reply: Thanks for your kindly suggestions! We have revised the title of Table 1 to “The physical property of 55 wasp mastoparans according to different subfamilies and species studied”.
7-Between lines 145 and 146 and 398…. please replace molecular weight by molecular mass. Although the literature accepts the term molecular weight, this is wrong. The correct one is molecular mass. Just for example, there is mass spectrometry and not weight spectrometry.
Reply: Thank you, we have replaced “molecular weight” to “relative molecular mass” in line 137 and 405.
8- In the line 85 please correct and separate the words …. Mastoparanshad15 amino acids
Reply: We are sorry for our wrong descriptions, we have corrected “Mastoparanshad15 amino acids” to “Mastoparans had15 amino acids” in line 80 of revised manuscript.
9- The same for line 96 …. among the55 mastoparan
Reply: We are sorry for our wrong descriptions, we have corrected “among the55 mastoparans” to “among the 55 mastoparan” in line 91 of revised manuscript.
10- Also why not place Table 1 before figure 2 in the text. It seems clear to me that Table 1 complements the data shown in figure 1 and thus more logical data would be shown.
Reply: Thanks for your professional suggestions! We have place Table 1 before Figure 2 in our revised manuscript.
11- About figure 2, line 136, I would change title. …from Subfamily classification of mastoparan family peptides from wasp venoms … to …. Classification of mastoparan family peptides from wasp venoms according different subfamilies.
Reply: Thanks for your kindly suggestions! We have changed the “Subfamily classification of mastoparan family peptides from wasp venoms” to “Classification of mastoparan family peptides from wasp venoms according to different wasp subfamilies.” in lines 140 of revised manuscript.
12- In the lines 136 and 137 … The sequence alignments of mastoparans… I would put …the amino acid sequence alignments …to differentiate from nucleotides alignments!
Reply: We are sorry for our incorrect descriptions, and we have changed all the “sequence alignments” to “the amino acid sequence alignments” in lines 141 to 143 of revised manuscript.
13- Still regarding Figure 2, the authors forgot to write the meaning of the different colors for the amino residues for letters A, C, D, E, and F. Would it be their classifications according to polarity, charges and hydrophobicity ?
Reply: We are sorry for our unclear descriptions. Actually, the color of different amino residues has no specific meaning, it is software that automatically labels different colors for aesthetic reasons in our manuscript.
14- In the line 152 the authors wrote …. and some scientists called it Mastoparan-L. Do the authors know why this molecule is called mastoparan L? why the letter L? an explanation for this could be write in the text.
Reply: We are sorry for our unclear descriptions. Mastoparan L was identified in Vespula lewisii venom in 1967, and the letter L is short for Lewisii. Mastoparan L is the first mastoparan family peptide, therefore, some scientists also call it Mastoparan shortly.
15- The sentence between lines 176 to 179 is repetitive and has been written previously, may be removed, or is incomplete in this part of the text. …..Fifty-five mastoparans were identified in five wasp families, including Eumenidae (13 mastoparans), Vespidae (22 mastoparans), Polistidae (9 mastoparans), Polybiidae (10 mastoparans) and Ropalidiidae (1 mastoparan).
Reply: Thanks for your kindly suggestions! We have deleted the lines 176 to 179 in our revised manuscript.
16- Between lines 185 to 187 the authors wrote …. These results suggest that Polistidae venom may be more allergically reactive and toxic than venom from the other four wasp families, while Eumenidae venom may be the least toxic. In my opinion the authors have data to discuss mast cell degranulating activities arising from mastoparans. But there are other toxins in the venoms and this conclusion may be speculative in terms of the allergenic potential of the studied species. Also this sentence would look better in the discussion.
Reply:we are very sorry for our ambiguous description. And we have removed the description “These results suggest that Polistidae venom may be more allergically reactive and toxic than venom from the other four wasp families, while Eumenidae venom may be the least toxic”, and added a new sentence “These results suggest that the number of mastoparans with degranulation activity in Polistidae venom is much higher than that in other four wasp families, while that from Eumenidae venom were lowest.” in lines 186 to 188 of revised manuscript.
17- In the legend of figure 3, Line 194, author could invert first Mastoporan (L) and then EpVP2b accordingly to the figures shown or invert the figures E and F.
Reply: We are sorry for our incorrect descriptions, and we have inverted the order of EpVP2b and Mastoparan(L) in figure legend in line 195 of revised manuscript.
18- Regarding Table 2, would it be good if the authors explained why they tested mast cell degranulation at 80 μg/mL? Any reason for not to test 100 μg/mL or 50 μg/mL?
Reply: we are sorry for our unclear description, In the preliminary experiment, we found that 80 μg/mL is the lowest dose of mastoparan that reaching 100% degranulation rate, therefore, we list the degranulation rate of each mastoparan at 80 μg/mL in Table 2 , and we add an interpretation: “We used the 50% effective concentration (EC50) and degranulation rate at 80 μg/mL (the lowest dose of mastoparans that reach to 100% degranulation rate) as two indexes to evaluate the degranulation activity of mastoparans in RBL-2H3 cells.” in lines 166 to 167 of revised manuscript.
19- Also on Table 2, as shown, it seems that the EC50 (μg/mL,) EC50 (μM) and R2 data were obtained from analyzes with RBL-2H3 cells. Is this interpretation correct? I think the table could be improved and clearer about this.
Reply: Thanks for your kindly suggestions! We have modified Table 2 and separated the degranulation rate at 80 μg/mL of RBL-2H3 and P815 cells.
20- I also would change the title of Table2 . from … The statistical of degranulation activity of 55 mastoparans in RBL-2H3 and P815 cells ….to … The degranulatory activity of mastoparans from wasp venoms on RBL-2H3 and P815 mast cell lines.
Reply: Thanks for your kindly suggestions! We have changed the title of Table 2 from “The statistical of degranulation activity of 55 mastoparans in RBL-2H3 and P815 cells” to “The degranulatory activity of mastoparans from wasp venoms on RBL-2H3 and P815 mast cell lines” in our revised manuscript.
21- Still on Table 2, in my opinion it should be shown in the text before figures 3 and 4, since it describes the same data shown for some mastoparans in figure 3, and additionally, figure 4 is calculated from data obtained in Table 2.
Reply: Thanks for your professional suggestions! We have transferred the Table 2 before Figure 3 and 4 in our revised manuscript.
22- Between lines 209 to 213 …. The difference in degranulation activity among these mastoparans from the same wasp may be related to their relative contents in wasp venom. For example, the content of Protopolybia-MPIII was significantly higher than that of Protopolybia-MPI/MPII in Protopolybiaexigua[14], and the contents of Pm-R1 and Pm-R3 were comparable and significantly higher than that of Pm-R2 in Polistes rothneyiiwatai venom[15]. Two possibilities: 1- this interpretation should come in the discussion. 2- more or less concentrated toxins in different venoms does not mean that these toxins are more or less active. They can be part of a family that act synergistically or summarily in the same biological event.
Reply: Thanks for your professional suggestions! We have transferred whole paragraph between lines 203 to 218 to discussion of lines 322 to 338.
23-Lines 213 to 216 …. These results suggest that one or two mastoparans in wasp venom may be the main active ingredient(s) that induce(s) mast cell degranulation, and other mastoparans may have different biological functions. Again in my opinion this interpretation of the results should come in the discussion. Also this sentence is controversial since the actions can be summative, different members of a Family of a same toxin in the same venom can also mean that this venom is adapted to act in several cellular conditions, different membranes, different pHs, humidity, resistance to inhibitors, among other explanations. I suggest to authors to complete this sentence!
Reply: Thanks for your kindly suggestions! We have transferred lines 213 to 216 to discussion, and we changed it to the “The degranulation activity of some mastoparans derived from the same wasp venom differs greatly; for example, Protopolybia-MPI/MPII/MPIII were identified in Protopolybia exigua, and the EC50 values were 116.76±38.30 μM, 77.85±9.08 μM and 22.47±2.96 μM, respectively. Pm-R1, Pm-R2, and Pm-R3 were identified in Polistes rothneyiiwatai, and the EC50 values of Pm-R1, Pm-R2, and Pm-R3 in inducing RBL-2H3 cell degranulation were 22.95±4.77 μM, 94.79±73.52 μM, and 7.60±0.76 μM, respectively (Table 2). Interestingly, the content of Protopolybia-MPIII was significantly higher than that of Protopolybia-MPI/MPII in Protopolybia exigua[14], and the contents of Pm-R1 and Pm-R3 were comparable and significantly higher than that of Pm-R2 in Polistes rothneyiiwatai venom[15], which suggested the difference in degranulation activity among these mastoparans from the same wasp may be related to their relative contents in wasp venom, and the role of different mastoparans in the same wasp venom played in degranulation activity needs further investigation.” in lines 323 to 336 of revised manuscript.
24- Lines 216 to 219 … In some wasp venom, we also found that not all mastoparans induced mast cell degranulation; for example, there were two mastoparans (Mastoparan-VB1 and Mastoparan-VB2) identified in Vespa bicolor, but neither of them could induce the degranulation of RBL-2H3 cells, which may be related to the toxicity of the wasps themselves. Once again, more suitable phrase for the discussion of the text. In addition, not causing mast cell degranulation does not mean absence of activities of these molecules. These mastoparans can bind to mast cells and cause activation of intracellular signals responsible for the release of chemokines, among several other signals. Here additional experiments could be done as an immunofluorescence reaction, to show whether these toxins bind to the surface of the cells, originating, for example, formation of microvesicles on the surface or diverse intracellular signals.
Reply: Thanks for your kindly suggestions! We have changed the sentence “which may be related to the toxicity of the wasps themselves.” to “which suggest that these mastoparans may have other biological activity and need further investigations” in line 337. Actually, the activation of mast cell by mastoparans could promote the release of some cytokines and chemokines which related with the recruitment of neutrophils or other cells, it is also one of the important research directions for us now.
25- Lines 225 to 226 …The authors wrote …. These results suggest that the mechanism underlying mast cell degranulation induced by mastoparans from different species may be different (Table 2). The degranulation mechanisms may be the same! What can vary are the receptors for the peptides, which can be expressed at different concentrations. Also the lipid composition of the cell membrane in different cells is different, thus influencing the same intracellular signaling. In addition to the cytoskeleton, inactive genes, among other examples of the intracellular machine that can participate in the same cell signaling. I suggest for the authors complete this phrase!
Reply: Thanks for your kindly suggestions! We have changed the sentence “These results suggest that the mechanism underlying mast cell degranulation induced by mastoparans from different species may be different (Table 2).” to “These results suggest that the degranulation activity of mastoparans might be varies in different cells (Table 2).” in line 210 and 211 of revised manuscript.
26- Regarding the data shown in figures 5, 6, 7 and 8 since the authors have all the mastoparans synthesized and purified, it would be interesting to carry out NMR analyses, and to determine the real structural organizations, if not all, of at least some of these peptides. This would be more reliable than performing comparative analyzes with mastoparan (L).
Reply: Thanks for your kindly suggestions! NMR analyses of the structure of mastoparans is a very important work to study the structure-activity relationship, and we are planning to work on it in our future research with a proper collaborator.
27- For legends of figures 5, 6, 7 and 8 complete…. Multiple amino acid sequence alignment analysis…..
Reply: We are sorry for our incorrect descriptions, we have corrected all the “Multiple alignment” to “Multiple amino acid sequence alignment” in the legends of figure 5,6,7,8 in revised manuscript.
28- Also in the analysis of amino acid alignment in figures 5, 6, 7 and 8 the amino acid residues were marked with different colors (black, pink, white and green), some special reason for this since there is no information in the text or subtitles?
Reply: We are sorry for our unclear descriptions. We have added a sentence “Amino acid residues were marked with dark blue (100% identity), pink (>75% identity), light blue (>50% identity) and blank (< 50% identity);” in the legends of figure 5,6,7,8 in revised manuscript.
29- Even through comparative analyses, using mastoparan (L) as a reference, the authors could carry out molecular dynamic analyses, which could bring more reliability to the hypotheses raised for the structure/function relationships.
Reply: Thanks for your kindly suggestions! And as you have suggested about NMR, we will carry out the molecular dynamic analyses of mastoparans in our future work with the collaborator.
30- The data shown in Table 3 and figure 9 are quite interesting and bring competitiveness to the text. I only suggest for other studies, that the authors carry out molecular dynamic studies comparing the wild forms with the mutated ones, using models of artificial membranes for instance. These data may strengthen the hypotheses indicated by the mutations.
Reply: Thanks for your kindly suggestions! And as you have suggested about NMR, we will carry out the molecular dynamic analyses of mastoparans in our future work with the collaborator.
31- The text written between lines 308 to 313 could come at the end of the discussion as it is interpretive and conclusive.
Reply: Thanks for your kindly suggestions! And we have transferred lines 308 to 313 to the end of discussion in revised manuscript.
32- In the line 315 ….Wasp venom is rich in various bioactive substances, such as peptides and proteins. Please cite a reference after the text.31
Reply: Thanks for your kindly suggestions! We have cited a reference at line 300 in revised manuscript.
33- In the lines 320 and 321 ….Since the discovery of the first mastoparan in wasp venom in 1979, scientists have identified a variety of mastoparans from different wasp venoms. Please cite a reference after the text.32
Reply: Thanks for your kindly suggestions! We have cited a reference at line 305 in revised manuscript.
34- In the lines 333 to 335 …. The degree of allergic reactions to wasp stings depends on multiple factors, such as the number of stings, the patients’ constitution and the wasp species (the toxicity of venom). Please cite a reference after the text.33
Reply: Thanks for your kindly suggestions! We have cited a reference at line 319 in revised manuscript.
35- In the lines 345 to 346 … In addition, 55 mastoparans can be divided into 4 subfamilies. This information was already written in other part of the text. Please remove it from here.
Reply: We are sorry for our unclear descriptions. And we have removed this sentence in the revised manuscript.
36- The text in the lines 355 to 357 … Delazari dos Santos et al found that Protopolybia-MPIII can cross the cell membrane into the plasma and contribute to the degranulation of mast cells [18], but the role of mastoparan hydrophobicity in membrane crossing needs further investigation. This part of the text is a clear justification to development of Molecular Dynamic analysis for mastoparans and cell membrane interactions!
Reply: Thanks for your kindly suggestions! We will explore the interactions between mastoparans and cell membrane by molecular dynamic analysis in the future.
37- In the line 365 the authors wrote …. Alessadra V et al. also found that …. The right citation for reference 20 is Cerovsky et al., Please check all references indicated along the text to avoid mistakes.
Reply: We are sorry for our mistake. And we have corrected the right reference at line 366 and 369 in revised manuscript.
38- In the line 380 the authors wrote … Delazari dos Santos L et al … The right citation for reference 18 is Goncalves et al.,
Reply: We are sorry for our mistake. And we have cited the right reference at line 355 and 356 in revised manuscript.
39- In the lines 400 and 401 … Multiple alignment of wasp mastoparans amino acid sequences and homology trees were analyzed using DNAMAN software. Please indicate a reference for this methodology.
Reply: Thank you! And we have cited the reference in the revised manuscript.
40- Line 400…. Multiple amino acid alignment …. Include the word amino acid
Reply: We are sorry for our mistake. And we have corrected the “Multiple alignment” to “Multiple amino acid alignment” at line 411 in revised manuscript.
41- Authors must include in the Material and Methods the details of how methodology they used to obtain mutated isoforms of mastoparans.
Reply: We are sorry for our unclear descriptions. And we have added the methodology we used to obtain the mutated isoforms of Protopolybia-MPIII in line 413 and 414 in the revised manuscript.
42- Also transcriptome analysis.
Reply: We are sorry for our unclear descriptions. Considering that the results of transcriptome sequencing of venom glands of Vespa basalis and Vespa nigrithorax in the Qinba Mountain area are unpublished, we removed this part.
43- Throughout the text, the authors describe the degranulatory potentials of different mastoparans from wasp venoms on two mast cell lineages, under culture conditions. I ask if in any situation, animal mast cells were tested, for example from the mesentery of mice. This is a simple procedure where the mesentery of mice is collected and stretched over a plate with paraffin for support, and then exposed to the toxins studied at different concentrations and times, and then the mesentery is stained with toluidine blue and observed under light microscopy. Alternatively, animals could be treated in vivo and collected their mesentery to study mast cell degranulation.
Reply: Thanks for your kindly suggestions! We have not tested the animal mast cells so far, and we could carry it out on animal mast cells in vitro and in vivo in our future work.
44- Of course mastoparans were initially studied as mast cell degranulators. But I missed analyzes on other cellular events, such as the inflammatory response in animals, or effects on other cells co-resident with mast cells, such as fibroblasts and endothelium of blood vessels. These could, although for other manuscripts, bring great improving in the area.
Reply: Thanks for your kindly suggestions! It is highly consistent with our research directions.
45- Finally, in the legend of supplementary figure 1, the authors should indicate the meanings of the green and pink colors in the amino acid residues.
Reply: We are sorry for our unclear descriptions. We have added a sentence “Amino acid residues were marked with dark blue (100% identity), pink (>75% identity), light blue (>50% identity) and blank (< 50% identity);” in the legends of supplementary figure.
Best Regards.

Round 2
Reviewer 3 Report
After careful reading of the text of the revised manuscript, it is my opinion that the authors made changes that have made the text and data presented more attractive and scientifically more accurate. Also in the response letter to the reviewer, the authors were clear and accepted most of the suggestions indicated. In my opinion this revised version achieves the scientific rigor demanded by TOXINS, is better than the original version and can be published by the Editorial Board. Congratulations to the authors.
Author Response
After careful reading of the text of the revised manuscript, it is my opinion that the authors made changes that have made the text and data presented more attractive and scientifically more accurate. Also in the response letter to the reviewer, the authors were clear and accepted most of the suggestions indicated. In my opinion this revised version achieves the scientific rigor demanded by TOXINS, is better than the original version and can be published by the Editorial Board. Congratulations to the authors.
Reply: Thanks for reviewer's kindly comments!
